# Central Stimulatory Effect of Kynurenic Acid on BDNF-TrkB Signaling and BER Enzymatic Activity in the Hippocampal CA1 Field in Sheep

**DOI:** 10.3390/ijms24010136

**Published:** 2022-12-21

**Authors:** Katarzyna Roszkowicz-Ostrowska, Patrycja Młotkowska, Paweł Kowalczyk, Elżbieta Marciniak, Marcin Barszcz, Tomasz Misztal

**Affiliations:** The Kielanowski Institute of Animal Physiology and Nutrition, Polish Academy of Sciences, Instytucka 3, 05-110 Jabłonna, Poland

**Keywords:** kynurenic acid, BDNF, BER pathway, DNA glycosylases, hippocampus, sheep

## Abstract

Deficiency of neurotrophic factors and oxidative DNA damage are common causes of many neurodegenerative diseases. Recently, the importance of kynurenic acid (KYNA), an active metabolite of tryptophan, has increased as a neuroprotective molecule in the brain. Therefore, the present study tested the hypothesis that centrally acting KYNA would positively affect: (1) brain-derived neurotrophic factor (BDNF)-tyrosine receptor kinase B (TrkB) signaling and (2) selected base excision repair (BER) pathway enzymes activities in the hippocampal CA1 field in sheep. Both lower (20 μg in total) and higher (100 μg in total) doses of KYNA infused into the third brain ventricle differentially increased the abundance of BDNF and TrkB mRNA in the CA1 field; additionally, the higher dose increased BDNF tissue concentration. The lower dose of KYNA increased mRNA expression for 8-oxoguanine glycosylase (OGG1), N-methylpurine DNA glycosylase (MPG), and thymine DNA glycosylase and stimulated the repair of 1,N6-ethenodeoxyadenosine and 3,N4-ethenodeoxy-cytosine as determined by the excision efficiency of lesioned nucleobases. The higher dose increased the abundance of OGG1 and MPG transcripts, however, its stimulatory effect on repair activity was less pronounced in all cases compared to the lower dose. The increased level of AP-endonuclease mRNA expression was dose-dependent. In conclusion, the potential neurotrophic and neuroprotective effects of KYNA in brain cells may involve stimulation of the BDNF-TrkB and BER pathways.

## 1. Introduction

Many neurodegenerative diseases associated with learning, memory, and behavior disorders, such as Alzheimer’s disease (AD), multiple sclerosis, Parkinson’s disease (PD)- or HIV-associated dementia and epilepsy are characterized by hippocampal atrophy, which manifests as a selective reduction in neuronal number and synaptic integrity [1,2]. Both pathology studies and high-resolution volumetric imaging performed in subjects in the early stages of AD revealed preferential degeneration of the hippocampal *cornu ammonis* (CA1) field [3,4]. Moreover, pathophysiological processes within CA1 neurons are believed to be responsible for the decrease in hippocampal volume in individuals with schizophrenia or suffering from other psychosis [5,6]. Regardless of the different etiology and pathogenesis of neurodegenerative diseases, the pathology of CA1 neurons appears to share several common features, such as changes in glutamatergic transmission, leading to excess glutamate and activation of locally resident microglia [7,8]. According to Small et al. [9], the preference for CA1 field neurodegeneration is due to the higher expression of N-methyl-D-aspartate (NMDA) receptors and the resulting vulnerability to glutamate-mediated neurotoxicity. On the other hand, activation of the innate immune system was shown to induce oxidative damage to neuronal membranes and degeneration of the dendritic system of CA1 pyramidal neurons [10,11]. In addition, changes in the neural hippocampal architecture may be caused by a loss of trophic support, as manifested by a decrease in the level or activity of neurotrophins [12].

Neurotrophins are a group of proteins secreted in the CNS, that include the most studied brain-derived neurotrophic factor (BDNF), as well as nerve growth factor (NGF), neurotrophin-3 (NT3), and neurotrophin-4 (NT4). They play major roles in synaptic transmission, neurogenesis, and neuronal plasticity, as well as regulate many different cellular processes involved in the development and maintenance of normal brain function, and those related to learning and memory [13,14,15]. Lack of functional *NGF, BDNF,* and *NT3* genes results in severe neuronal deficits and early postnatal death [16]. In particular, deficits in BDNF signaling have been reported to contribute to the pathogenesis of several major diseases, such as Huntington’s disease, AD, depression, schizophrenia, and mental or anxiety disorders [17,18]. The biological effects of neurotrophins are exerted by the activation of specific receptors, belonging to the family of tyrosine protein kinase receptors (TrkA, TrkB, and TrkC), as well as the low-affinity receptor p75NTR [13,14]. Among them, TrkB shows a strong binding preference for BDNF and is highly expressed in the CNS by both neurons and astrocytes [19].

Due to the high metabolic activity, the rate of reactive oxygen species (ROS) production in neuronal cells is higher than in other cell types. When protective antioxidant systems are disturbed, abnormal ROS accumulation causes oxidative and nitrosative damage to proteins, lipids, and DNA molecules [20,21]. The latter are the most dangerous for the body since DNA lesions can induce mutagenic chromosomal changes and/or hinder essential biological processes, namely transcription or replication [21,22]. In addition, DNA damage may trigger apoptosis, which occurs in newly generated neurons during the development of the nervous system or adult neurogenesis [23,24]. In order to maintain the proper functioning of cells, various DNA repair mechanisms have evolved to prevent the accumulation of deleterious mutations, such as the base excision repair (BER) pathway, which is responsible for the removal of damaged nucleobases [25]. The primary enzymes of the BER system include damage-specific DNA glycosylases, which are responsible for the recognition and removal of altered bases and assisting AP-endonuclease (APE1), which in turn cleaves at the apurinic/apyrimidinic (AP) site created after removal of the damaged base [26].

Kynurenic acid (KYNA) is an active tryptophan metabolite, mainly formed as a result of L-kynurenine transamination by kynurenine aminotransferase in the CNS and various peripheral mammalian tissues [27,28]. Significant quantities of KYNA in the brain are produced in glial cells, from where it is released into the extracellular space by passive diffusion [28]. KYNA is considered a natural neuromodulator, as it can interact with NMDA and nicotinic receptors, inhibiting glutamatergic and cholinergic transmission [29,30] and changing extracellular concentrations of dopamine and *γ*-aminobutyric acid (GABA) [31,32]. All these receptors and neurotransmitters have been shown to mediate several important processes in the CNS, including neurodevelopment, plasticity, cognition, behavior, and memory formation. On the other hand, the overactivation of glutamate receptors may induce, at least partly, neurodegenerative processes leading to brain damage, which is why interest has focused on kynurenine pathway manipulation [33]. Moreover, KYNA has also been shown to act as a ligand capable of activating G-protein-coupled receptor 35 (GPR35), thereby reducing the release of proinflammatory cytokines, such as tumor necrosis factor-α [34,35]. In addition, evidence points to the antioxidant and ROS-scavenging properties of KYNA—it has been shown to scavenge various ROS and decrease the level of important oxidative damage markers produced by different pro-oxidants in tissue preparations [36]. Considering the contribution of ROS to the development of neurodegenerative diseases and the aging process in the brain [37], as well as the endogenous nature of KYNA associated with various CNS pathologies [38], demonstrating the neurotrophic and/or neuroprotective effects exerted by this tryptophan metabolite is still of major importance. The relationships between KYNA and the BDNF-TrkB signaling pathway, as well as KYNA and front-line DNA repair (BER pathway) enzymes in the CNS, are not well understood. Therefore, the present study tested the hypothesis that centrally acting KYNA would positively affect: (1) BDNF and TrkB mRNA expression and BDNF tissue concentration, as well as, (2) mRNA expression of selected BER pathway enzymes and their excision efficiency of damaged nucleobases in the hippocampal CA1 field in sheep.

## 2. Results

### 2.1. BDNF mRNA Expression, Tissue Concentration, and TrkB Receptor Expression

Differences in the relative abundance of the *BDNF* gene transcript and BDNF concentration in the hippocampal CA1 field between individual groups of sheep are shown in Figure 1. Both the lower and higher KYNA doses increased the relative abundance of BDNF transcript (*p* < 0.05 and *p* < 0.001, respectively) in comparison with the control group. Moreover, the level of BDNF mRNA in the CA1 field was higher (*p* < 0.001) in high dose-infused sheep compared to the animals that received the lower dose. The higher dose of KYNA also increased (*p* < 0.05) BDNF concentration in the CA1 field compared to controls (757.14 ± 81.89, 906.60 ± 131.41, and 1302.10 ± 179.13 pg/mg protein, for the control, lower, and higher doses, respectively). TrkB receptor mRNA expression increased in response to the lower (*p* < 0.05) and higher (*p* < 0.001) KYNA doses compared to the control group, and the effect of KYNA was more pronounced at its higher dose (*p* < 0.001) (Figure 2).

### 2.2. mRNA Expression and Activity of DNA Glycosylases

The transcripts of all tested DNA glycosylases (OGG1, MPG, and TDG) and APE1 were detected in the CA1 field of the hippocampus. The relative abundance of these transcripts in all treatment groups is shown in Figure 3. A significant increase (*p* < 0.01–*p* < 0.001) in mRNA expression for all glycosylases in the hippocampal CA1 field was observed after administration of the lower KYNA dose in comparison with controls. The response in terms of transcription of glycosylases to the higher dose was variable: KYNA stimulated the expression of OGG1 (*p* < 0.001) and MPG mRNA (*p* < 0.01), but had no effect on TDG mRNA expression compared to controls. Moreover, the abundance of OGG1 and MPG transcripts in the CA1 field did not differ between sheep treated with the lower and higher KYNA doses, while TDG mRNA levels in the lower dose-infused sheep were higher (*p* < 0.01) than in the animals infused with the higher dose. The level of APE1 mRNA expression was dose-dependent: both lower and higher doses increased (*p* < 0.01) transcript abundance in the CA1 field compared to the control animals, but APE1 mRNA expression in sheep infused with the higher dose was higher (*p* < 0.01) compared to the animals infused with the lower dose.

The repair activity of glycosylases for individually modified nucleobases differed between the treatment groups (Figure 4). Neither of the two doses of KYNA significantly influenced the efficiency of 8-oxoG excision compared to the control group (6.40 ± 0.06, 7.17 ± 0.17, and 6.08 ± 0.31 fmol/mg protein/h, for the control, lower, and higher doses, respectively). Interestingly, the rate of 8-oxoG excision in the higher-dose group was lower (*p* < 0.01) than in the group treated with the lower dose. Both doses of KYNA increased the excision efficiency for εA (*p* < 0.001) and εC (*p* < 0.001) compared to controls; however, the effect of the higher dose was less pronounced in both cases (*p* < 0.001) than for the lower dose (εA: 2.96 ± 0.05, 6.16 ± 0.12, and 5.43 ± 0.08 fmol/mg protein/h, for the control, lower, and higher doses, respectively, and εC: 2.45 ± 0.06, 5.85 ± 0.15, and 4.66 ± 0.20 fmol/mg protein/h, for the control, lower, and higher doses, respectively).

## 3. Discussion

The ventricular system of the sheep’s brain, as in humans, includes the lateral ventricles, third ventricle, cerebral aqueduct, and fourth ventricle, which are filled with cerebrospinal fluid (CSF). According to the classical model, CSF production takes place mainly at the choroid plexuses located in the ventricles and their ependymal layer [39]. Therefore, the CSF flow is directed caudally through the ventricles and the subarachnoid space toward the arachnoid villi, where it is absorbed into the venous blood. The existence of the laminar flow of CSF in a thin layer along the ventricular walls, driven by the ependymal ciliary movements, may contribute to the mixing of CSF inside the ventricular system [39]. More recent experimental and clinical data indicate that CSF is also formed and reabsorbed across the walls of CNS blood capillaries [40]. Furthermore, CSF circulation seems much more complex: a combination of directed bulk flow and pulsatile flow, which is a bidirectional movement in an upward (cranial) and downward direction (caudal) along the spinal cord, and in varying directions in the brain. Rodent studies demonstrated that drugs could very well be transported throughout the entire brain, following intrathecal application [41]. Considering the ventricular system organization, the possibility of multidirectional CSF circulation, as well as the immediate vicinity of the paired structure of the hippocampus to the lateral ventricles, the test substance was infused to the IIIv in the present study. It has previously been shown that a slow and intermittent infusion regimen can elicit clear biological responses to a given compound in both close and more distant brain structures without reducing receptor sensitivity [42,43].

As demonstrated, both BDNF mRNA expression and BDNF concentration in the hippocampal CA1 field increased in response to the central infusion of KYNA. While gene transcript levels increased with a rising KYNA dose, a significant increase in protein accumulation in the examined tissue was observed only at the higher dose. Although BDNF is synthesized in neurons as a pre-pro neurotrophin, which is subsequently cleaved into pro-BDNF [44], our tissue concentration measurements for this compound concerned the mature form, stored in secretory vesicles and released locally. It has been established that BDNF transcription and release are mainly stimulated by excitatory synaptic activity, especially involving ionotropic glutamatergic NMDA receptors (NMDAR) and calcium-mediated pathways [45]. While KYNA is a known NMDAR antagonist, it may interact with different receptor binding sites (glutamate and/or glycine) and also antagonize other receptor types, including α-amino-3-hydroxy-5-methyl-4-isoxazolepropionic acid (AMPA) and kainate receptors [29,30,34,35]. An earlier study by Liu and Moghaddam [46] demonstrated that basal glutamate and aspartate outflow in the hippocampus of freely moving rats was increased in a dose-dependent manner by local application of the NMDAR or non-NMDAR antagonists. Research on antidepressant factors showed that ketamine and other NMDAR antagonists produced fast-acting behavioral antidepressant-like effects in murine models that were associated with rapid BDNF synthesis [47]. Moreover, these fast-acting antidepressant-like effects relied on enhanced neurotransmission after NMDAR-antagonist-induced plasticity and occurred at rest. A more recent study confirmed that NMDAR antagonists led to the disinhibition of glutamate transmission, causing a glutamate surge. Glutamate influx increased BDNF release and promoted the expression of synaptic proteins [48]. On the other hand, the involvement of AMPA receptors in the ketamine-induced upregulation of hippocampal BDNF was demonstrated in rats [49]. According to Maroni et al. [35], KYNA could produce a range of responses throughout the CNS due to the differential signal transduction mechanisms and downstream signaling cascades.

A similar dose-dependent response was also observed for mRNA expression of the TrkB receptor that transmits BDNF signaling. BDNF binding to TrkB results in receptor dimerization and trans-autophosphorylation, which in turn engages multiple kinase signaling cascades, including PI3 kinase, Akt, ERKs, and protein kinase C [50]. It is well established that BDNF-TrkB signaling plays a crucial role in neuronal plasticity and differentiation in the postnatal hippocampus, and increasing BDNF levels in the hippocampal CA1 field may have beneficial effects on learning and memory functions [13,14,51]. In this aspect, BDNF is the main factor triggering an increase in dendritic spine density and the total length of apical dendrites within the CA1 field [51,52]. It also participates in the regulation of synaptic excitatory transmission, specifically by enhancing the phosphorylation of NMDAR subunits [53]. In addition, BDNF can maintain the functional efficiency of neurons by stimulating antioxidant and DNA repair mechanisms. Previous studies showed that exogenous treatment of cultured hippocampal neurons with this neurotrophin induced an increase in antioxidant enzyme activities, thereby suppressing cellular ROS accumulation [54]. Moreover, BDNF could stimulate neuronal DNA repair by enhancing APE1 expression, a key base excision enzyme in the DNA repair pathway [55].

Considering the protection of CNS cells against oxidative stress, especially at the molecular level, this appears to be the first report demonstrating the functional relationship between KYNA and the enzymatic activity of the BER pathway in hippocampal tissue. More specifically, increased KYNA concentration in the sheep’s CNS enhanced the *OGG1, MPG, TDG,* and *APE1* transcript expression (except for the *TDG* response to the higher dose of KYNA), as well as, to some extent, the excision efficiency of damaged nucleobases in the hippocampal CA1 field. Selected glycosylases of the BER pathway identify and eliminate specific DNA base lesions caused either by exogenous mutagens or endogenously generated factors as a result of oxidative stress or lipid peroxidation [56]. The most pronounced increase in mRNA expression was recorded for OGG1 in response to both KYNA doses. This glycosylase is the main enzyme responsible for the removal of 8-oxoG, an oxidized form of guanine, which is considered one of the major endogenous mutagens, as it can form a relatively stable pair with adenine and cytosine in DNA during replication [56]. Uncontrolled 8-oxoG accumulation in both nuclear and mitochondrial DNA is a major cause of cancer and cell death, which is associated with neurodegenerative processes in the brain [57]. Since the formation of such a DNA base lesion is caused by ROS, the stimulation of OGG1 mRNA expression by KYNA is consistent with its antioxidant effects. This tryptophan metabolite has been previously shown to be effective in hydroxyl radical scavenging in various toxicity models with induced ROS production [36,58]. Therefore, it can be assumed that the presented lack of clear changes in the efficiency of 8-oxoG excision between the sheep groups could, at least partially, reflect the predominance of KYNA’s own scavenging properties, and thus lower oxidized guanine formation in DNA, or it could indicate a longer processing time of the accumulated OGG1 mRNA. The latter could result from a more complicated bifunctional nature of this enzyme, which exhibits both DNA glycosylase and AP-lyase activities [59]. Moreover, there is evidence that KYNA can also prevent lipid peroxidation (LPO) and the formation of harmful LPO-derived aldehydes [60]. The latter is responsible for the generation of ε-derivatives of DNA bases, such as εA and εC, which have as high mutagenic potential as 8-oxoG [61]. It has been established that both ε-bases are recognized and excised by mono-functional DNA glycosylases: MPG, which typically repairs alkylation-damaged DNA bases, and TDG, which is mainly implicated in the repair of deamination-derived DNA lesions [56]. To this end, our study showed increased expression of gene transcripts of both glycosylases, as well as the efficiency of ε-base excision in the hippocampal CA1 field in response to central KYNA infusion. Importantly, the lower excision efficiency obtained at the higher dose may confirm KYNA’s own antioxidant properties [60] and, consequently, a lower degree of nucleobase damage.

KYNA also positively affected the *APE1* gene expression in a dose-depending manner. This enzyme recognizes and cleaves AP sites formed by glycosylases, thereby limiting their effects [26]. As shown, the highest three-fold increase in transcript levels was induced by the higher dose of KYNA. Based on our data on the hippocampal BDNF level (an increase in response to the higher dose) and the literature [55], it can be assumed that there was a cumulative stimulatory effect of both factors: KYNA and BDNF. In general, the stimulation of gene expression of BER pathway enzymes and BDNF synthesis, following the transient activation of glutamate receptors can trigger the same mechanism, involving activation of cAMP response element-binding protein (CREB) [62]. CREB, which also mediates BDNF signaling, has been shown to be the major transcription factor associated with *APE1* gene expression [55]. Accordingly, KYNA alone or via locally released BDNF can contribute to antioxidant brain protection by enhancing the ability of neurons/glial cells to repair oxidative nucleobase modifications.

Although the present study has demonstrated the positive effects of KYNA on the CNS, the use of this tryptophan metabolite as a potential strategy for the treatment of neurodegenerative and/or mental diseases faces many limitations and controversies. Unlike its precursor L-kynurenine, KYNA crosses the blood-brain barrier only in a small amount, therefore peripheral KYNA is not expected to contribute significantly to brain pools [63]. Basal extracellular KYNA concentrations in the rodent brain are in the low nanomolar range, and in the human brain are 20–50 times higher [64,65]. It is believed that changes in cerebral kynurenine pathway activity could contribute to reduced neurogenesis and increased excitotoxicity in neurodegenerative disease. Reduced cerebral KYNA production was observed in AD and PD patients compared to age-matched control [65]. Therefore, deficiency of KYNA implicates the potential therapeutic value of increasing brain KYNA levels. On the other hand, elevated levels of KYNA in CSF have been observed in relapsing-remitting multiple sclerosis patients, which is a demyelinating disease [66]. Rodent studies showed that excessive and/or prolonged exposure to this compound (in the nanomolar or low micromolar range) may cause a spectrum of cognitive, memory, and motor deficits, including damage and loss of myelin [67,68,69]. In contrast to the above data, the selected dose of KYNA and the administration scheme presented in sheep a short-term effect without the expected injury of brain tissue. No animal was found to be neurologically impaired during the experimental procedure and no local foci of inflammation were found in the post-slaughter examination.

In conclusion, the present study demonstrated a central stimulatory effect of KYNA on BDNF-TrkB signaling and BER enzymatic activity in the hippocampal CA1 field in sheep. However, the obtained results should be approached with caution due to the conflicting results and adverse KYNA effects observed in other experimental models.

## 4. Materials and Methods

### 4.1. Animal Management

Eighteen Polish Longwool sheep (a breed showing reproductive seasonality), aged 1 year and weighing 55 ± 2 kg were used in the experiment. The animals were kept indoors at the Sheep Breeding Center of the Kielanowski Institute of Animal Physiology and Nutrition, Polish Academy of Sciences (Jablonna near Warsaw, Poland) under natural lighting conditions (52° N, 21° E). They were fed twice a day according to their physiological status. Animal nutrition was based on pelleted concentrate and hay in accordance with the recommendations of the National Research Institute of Animal Production (Krakow-Balice, Poland), and the National Institute for Agricultural Research (France) [70]. During the experimental period, the sheep were kept in individual pens, providing visual, olfactory, and tactile contact and free access to water and mineral licks.

### 4.2. Third Ventricle (IIIv) Cannulation

One month before the experiment, the sheep were surgically implanted with a cannula into the third ventricle (IIIv) of the brain (outer diameter—1.2 mm, position: frontal—31.0 mm), in accordance with the stereotaxic coordinate system for the sheep hypothalamus [71]. Implantation was performed under general anesthesia (xylazine: 40 mg/kg body mass, intravenously; xylapan and ketamine: 10–20 mg/kg body mass, intravenously; Bioketan, Vetoquinol Biowet, Pulawy, Poland) through a hole drilled in the skull, in accordance with the procedure described by Traczyk and Przekop [72]. A guide cannula was fixed to the skull with stainless steel screws and dental cement. The external opening of the canal was closed with a stainless steel cap. After surgery, the sheep were injected for four days with antibiotics (1 g streptomycin and 1,200,000 IU benzylpenicillin; Polfa, Poland) and analgesics (metamizole sodium: 50 mg/animal; Biovetalgin, Biowet Drwalew, Poland or meloxicam: 1.5 mg/animal; Metacam, Boehringer Ingelheim, Germany). The placement of the cannula in the IIIv was confirmed by CSF outflow during surgery and after slaughter. The sheep used in the present study had correctly located cannulas.

### 4.3. Experimental Design and Tissue Collection

The experiment was performed in March during the natural anestrous season for this breed of sheep. The animals were randomly divided into three groups (n = 6 each) and infused into the IIIv with Ringer-Locke solution (RLs, control) or with one of two doses of KYNA (Sigma Chemical Co., St Louis, MO, USA) dissolved in RLs. The treatment was performed in a series of four 30 min infusions, at 30 min intervals, from 10:00 to 14:00. KYNA doses (lower: 4 × 5 μg/60 μL/30 min, and higher: 4 × 25 μg/60 μL/30 min) were selected on the basis of scientific literature [33,35]. All infusions were performed using a BAS Bee microinjection pump (Bioanalytical Systems Inc., West Lafayette, IN, USA) and calibrated with 1.0 mL gas-tight syringes. During the treatments, sheep were kept in pairs in the experimental room in comfortable cages, where they could lie down and to which they had been previously adapted for three days. Immediately after the experiment, the sheep were slaughtered after prior pharmacological stunning (xylazine 0.2 mg/kg body mass and ketamine: 3 mg/kg body mass, intravenously) and the brains were rapidly removed from the skull. After the separation of the median eminence and cerebellum, each brain was sectioned sagittally into the cerebral hemispheres. The hippocampus was dissected from the medial part of the temporal lobe of the right hemisphere, starting from the floor of the lateral ventricle through ventral and dorsal parts, according to the sheep brain atlas [73]. Sections of approximately 2–3 mm in length were cut out from the CA1 field of the hippocampus. All tissue incisions were performed on sterile glass plates placed on ice, and subsequently, the collected structures were immediately frozen in liquid nitrogen and stored at −80 °C.

### 4.4. Analysis of Relative mRNA Abundance

Total RNA from CA1 tissues was isolated using the NucleoSpin RNA II kit (MachereyNagel, Düren, Germany) 4 × 4500, according to the manufacturer’s protocol. The concentration and purity of isolated RNA were quantified using a NanoDrop ND-1000 spectrophotometer (Thermo Fisher Scientific, Waltham, MA, USA). RNA integrity was electrophoretically verified on a 1.5% agarose gel stained with ethidium bromide. The TranScriba Kit (A&A Biotechnology, Gdynia, Poland) 10×1000 was used to synthesize cDNA according to the manufacturer’s instructions (1 µg of total RNA in a reaction volume of 20 µL). Quantitative polymerase chain reaction (qPCR) was performed using 5× HOT FIREPol^®^ EvaGreen qPCR Mix Plus (Solis BioDyne, Tartu, Estonia). The PCR amplification mix contained 2 µL of cDNA template, 1 µL of primers (0.5 µL each at 10 pmol/mL), 3 µL of PCR Master Mix buffer, and 9 µL of dd H_2_O. The reaction conditions were as follows: initial denaturation at 95 °C for 15 min, denaturation at 95 °C for 15 s, annealing at 60 °C for 20 s, and elongation at 72 °C for 20 s (40 cycles). Specific primers for determining the expression of *BDNF* and *TrkB* genes, 8-oxoguanine glycosylase (*OGG1*), N-methylpurine DNA glycosylase (*MPG*), thymine DNA glycosylase (*TDG*), and AP-endonuclease 1 (*APE1*) genes, as well as endogenous control genes: glyceraldehyde-3-phosphate dehydrogenase *(GAPDH)*, peptidylprolyl isomerase C *(PPIC)*, and 18S ribosomal RNA (*18S rRNA*), were designed using Primer3 software (The Whitehead Institute, Boston, MA, USA) and are listed in Table 1. Amplification specificity was further validated by electrophoresis of the obtained amplicons in a 2% agarose gel and visualized under a UV light camera. Data were analyzed with Rotor Gene 6000 v. 1.7 software (Qiagen, Hilden, Germany) using a comparative quantification option and Relative Expression Software Tool, based on the PCR efficiency correction algorithm developed by Pfaffl et al. [74,75]. The expression levels of the tested genes were normalized using geometric means of the expression of reference genes. Endogenous control genes were assayed in each sample to compensate for variations in cDNA concentration and PCR efficiency between individual tubes.

### 4.5. Tissue BDNF Concentration Analysis

Frozen hippocampal CA1 sections were mixed with radioimmunoprecipitation assay (RIPA) buffer (0.5 M Tris-HCl, pH 7.4, 1.5 M NaCl, 2.5% deoxycholic acid, 10% NP-40, 10 mM EDTA) (Merck, Darmstadt, Germany) at a ratio of 1:10 (tissue to reagents), and aprotinin as protease inhibitor (10 IU/mL, Sigma-Aldrich, Saint Louis, MO, USA). Each tissue sample was homogenized using a laboratory homogenizer and ceramic balls. After 30 min incubation on ice, the homogenates were centrifuged at 12,000× *g* for 10 min at 4 °C and the supernatants were then transferred to a new 1.5-mL Eppendorf tube and immediately stored at −80 °C for later use.

BDNF concentration in the homogenates was determined using the Biosensis Mature BDNF Rapid ELISA kit (BEK-2211, Biosensis Pty Ltd., Thebarton, Australia) according to the manufacturer’s protocol. Although originally scheduled for humans, mice, and rat studies, this ELISA kit can also be used for the quantification of mature BDNF in biological material obtained from other mammalian species, including sheep. It was shown to achieve reproducibility with intra- and interassay CVs of 1% and 5%, respectively, with a minimum detectable dose of BDNF less than 2 pg/mL. In addition, total protein concentration in tissue homogenates was analyzed spectrophotometrically by the Bradford method using the Bio-Rad Protein Assay Kit II (Bio-Rad, Hercules, CA, USA) according to the manufacturer’s instruction. BDNF concentration in each homogenate sample was determined as pg per mg of total protein.

### 4.6. Enzyme Repair Activity

Oligonucleotides (40-mers) containing a single 8-oxo-guanine (8-oxoG), 1,*N*^6^-ethenodeoxyadenine (εA) and 3,*N*^4^-ethenodeoxycytosine (εC) at position 20 in the 50-d (GCT ACC TAC CTA GCG ACC TXC GAC TGT CCC ACT GCT CGA)-30 sequence, where X indicated lesioned nucleobases, were obtained from Eurogentec Herstal (Herstal, Belgium) or Genset Oligos (Paris, France). Oligonucleotides were 32P-labeled at the 50-end by polynucleotide kinase at an excess of [32P]ATP (3000 Ci/mmol) (Amersham, Little Chalfont, UK). Radiolabeled oligomers were purified from unincorporated radioactive molecules using Micro Bio-Spin P-30 columns, as described by the manufacturer (Bio-Rad, Hercules, CA, USA). These oligomers were annealed at double-molar excess to complementary oligonucleotides containing T opposite εA, G opposite εC, or C opposite 8-oxoG. Complementary oligodeoxynucleotides were synthesized according to standard procedures using an Applied Biosystems synthesizer (Oligonucleotide Synthesis Laboratory, Institute of Biochemistry and Biophysics, Polish Academy of Sciences). The repair activity of BER pathway enzymes was determined based on the excision efficiency of damaged nucleobases (8-oxoG, εA, and εC) using the nicking assay, as described in detail in previous studies [43,76].

### 4.7. Statistical Analysis

Initially, all data were tested for normality using the Shapiro–Wilk normality test and then grouped into parametric and non-parametric groups. Tissue BDNF concentrations and excision efficiency of damaged nucleobases were analyzed using a one-way analysis of variance (STATISTICA, Stat Soft, Tulsa, OK, USA). The post-hoc Tukey test was performed after each analysis. Statistical evaluation of differences in the expression of OGG1, MPG, TDG, APE1, BDNF, and TrkB receptor mRNAs in the hippocampal CA1 field between treatment groups was carried out using non-parametric statistics, involving the Kruskal–Wallis test with multiple comparisons of mean ranks and the Mann–Whitney *U* test for individual groups. Differences were considered significant at *p* < 0.05, and all data are presented as mean ± standard error of the mean (SEM).

## Figures and Tables

**Figure 1 ijms-24-00136-f001:**
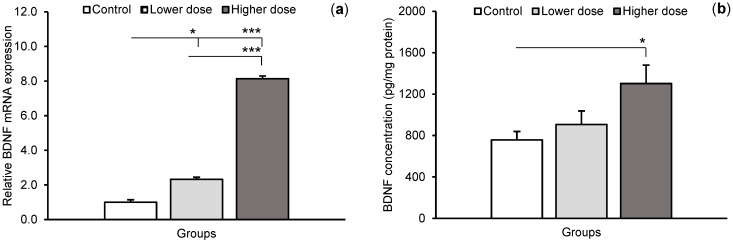
Brain-derived neurotrophic factor (BDNF) mRNA expression (**a**) and peptide concentration (**b**) (mean ± SEM) in the hippocampal CA1 field of sheep infused with the lower (20 μg in total) and higher (100 μg in total) dose of kynurenic acid into the third ventricle of the brain. Significance of differences: * *p* < 0.05, *** *p* < 0.001.

**Figure 2 ijms-24-00136-f002:**
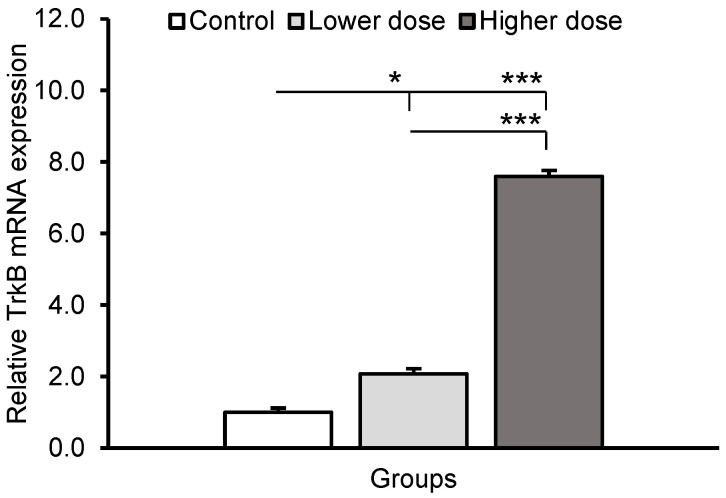
Tyrosine receptor kinase B (TrkB) mRNA expression (mean ± SEM) in the hippocampal CA1 field of sheep infused with the lower (20 μg in total) and higher (100 μg in total) dose of kynurenic acid into the third ventricle of the brain. Significance of differences: * *p* < 0.05, *** *p* < 0.001.

**Figure 3 ijms-24-00136-f003:**
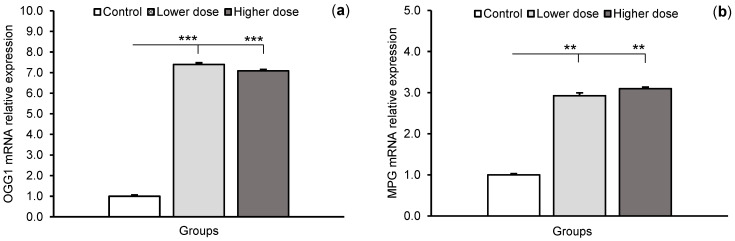
Relative mRNA expression (mean ± SEM) of 8-oxoguanine glycosylase (OGG1), (**a**) N-methylpurine-DNA glycosylase (MPG), (**b**) thymine-DNA glycosylase (TDG), (**c**) and AP-endonuclease 1 (APE1), (**d**) in the hippocampal CA1 field of sheep infused with the lower (20 μg in total) and higher (100 μg in total) dose of kynurenic acid into the third ventricle of the brain. Significance of differences: ** *p* < 0.01, *** *p* < 0.001.

**Figure 4 ijms-24-00136-f004:**
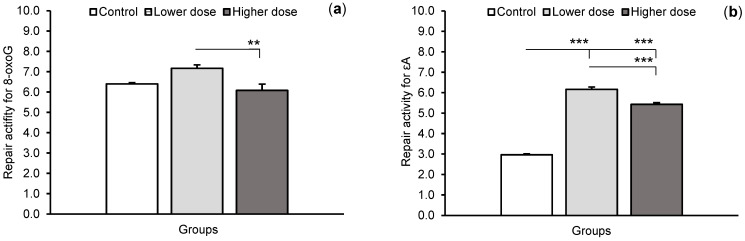
Repair activities (fmol/μg protein/h) for 8-oxoguanine (8-oxoG), (**a**) 1,N6-ethenoadenine (εA), (**b**) and 3,N4-ethenocytosine (εC), (**c**) in the hippocampal CA1 field of sheep infused with the lower (20 μg in total) and higher (100 μg in total) dose of kynurenic acid into the third ventricle of the brain. Significance of differences: ** *p* < 0.01, *** *p* < 0.001.

**Table 1 ijms-24-00136-t001:** Sequences of specific primers.

GENE	PRIMERS (5′–3′)	GENBANK ACC. NO.	AMPLICON SIZE
*BDNF*	F: CGTTGGCTGACACTTTTGAA R: CGCAGCATCCAGGTAATTTT	XM_012143442.1	188
*TRKB*	F: TGTCTGAGCTGATCCTGGTG R: TATCTGCAGGTTTGCCAGTG	XM_012117231.2	155
*OGG1*	F: CAGTCATAATAACAGTA R: AACCTCCTCTAAGCACTCAT	NC_040270.1/XM004018285	140
*MPG*	F: GCTGAGGGCCAGCCAACACCTGC R: CGCCCCTTTACCCACGGAGCCCA	NC_040275.1/XM027962018.1	140
*TDG*	F: ACACAGGATGCTGTGGGGCT R: TCCCTCGGCCTAGAATTTTC	NC_040254.1	120
*APE1*	F: TTAGACATTTGGTTGCC R: GGCACCAACAGGGCTAGCA	NC_040272.1	140
*GAPDH*	F: GGGTCATCATCTCTGCACCT R: GGTCATAAGTCCCTCCACGA	NM_001190390.1	131
*PPIC*	F: TGGAAAAGTCGTGCCCAAGA R: TGCTTATACCACCAGTGCCA	XM_004008676.1	158
*18S* *RRNA*	F: GCAATTATTCCCCCATGAACG R: GGGACTTAATCAACGCAAGC	NR_003286	115

BDNF: brain-derived neurotrophic factor, TrkB: tyrosine kinase receptor B, OGG1: 8-oxoguanine glycosylase, MPG: N-methylpurine-DNA glycosylase, TDG: thymine-DNA glycosylase, APE1: AP-endonuclease 1, GAPDH: glyceraldehyde-3-phosphate dehydrogenase, PPIC: peptidylprolyl isomerase C, 18S rRNA: 18S Ribosomal RNA, F: forward primer, R: reverse primer. The real-time PCR amplification efficiencies of target and reference genes were 96–101%.

## Data Availability

The datasets analyzed during the current study are available from the corresponding author upon reasonable request.

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
