# Peer review of "Central Stimulatory Effect of Kynurenic Acid on BDNF-TrkB Signaling and BER Enzymatic Activity in the Hippocampal CA1 Field in Sheep"

_ijms, 2022, doi:10.3390/ijms24010136_

Round 1

Reviewer 1 Report

The manuscript contains new and intriguing results indicating an unknown mechanism of action for kynurenic acid (KYNA). However,

·         The authors did not explain why KYNA was administered to the 3rd ventricle of the brain? Can the fluid administered into the 3rd ventricle of the brain come into contact with CA1 region of the hippocampus? Since the sheep is a rare object of research it is worth showing a scheme presenting  the localization of CA1 field of the hippocampus, hypothalamus (navigation point; line 278) and the 3rd ventricle of the brain with the direction of CSF flow marked. This issue should be adequately discussed.

·         It should be explained why KYNA was administered intermittently and not by continuous infusion (lines 295-296)?

·         It was reported that prolonged subdural infusion of KYNA resulted in myelin damage in rats (Dabrowski et al., 2015; DOI: 10.1371/journal.pone.0142598). Was a morphological examination of the CA1 field of the hippocampus performed to rule out damage? This issue should be thoroughly discussed.

·         Compare doses of KYNA used in manuscript with that utilized by Dabrowski et al., 2015 (DOI: 10.1371/journal.pone.0142598). Moreover, compare the concentration of KYNA administered intracerebrally in the presented manuscript to that known from experiments performed on cell lines in vitro. It has been repeatedly shown that KYNA applied in high concentrations markedly reduced viability of cells. Could such a phenomenon occur in the presented study on sheep?

·         The discussion as it currently stands does not present various possible scenarios. The authors suggest that KYNA increases the expression and content of the factors studied and that this is a positive effect. However, an increase in the expression and content of these factors also accompanies tissue insult or damage (see for review Brigadski and Leßmann 2020; DOI: 10.1007/s00441-020-03253-2).

·         Therefore, it is fundamental to analyze and discuss whether the noted changes are actually the result of KYNA's physiological effects or perhaps a response to excessively high toxic concentration of KYNA in the brain.

Minor points

·         Correct sentence (lines 175-177): “While KYNA is an NMDAR antagonist, it may interact with different receptor binding sites (glutamate and/or glycine), as well as activate other receptor types, including α-amino-3-hydroxy-5-methyl-4-isoxazolepropionic acid (AMPA) and kainate receptors [29,30,34,35]”. KYNA is an antagonist of both AMPA and kainite receptors.

·         Since KYNA increases BDNF in CA1 (this manuscript), and increased “BDNF levels in the hippocampal CA1 field may have beneficial effects on learning and memory functions [13,14,46] (lines 198-199), how to explain that it has been repeatedly reported that KYNA impairs memory?

·         Please, again analyze carefully the arguments used and cited. Not all of them are convincing.

Reviewer 2 Report

This is an interesting manuscript with a good perspective on the purpose and aim of the study.

In this paper the Authors highlighted the potential beneficial effect of KYNA on BDNF concentration and signaling and base excision repair enzymatic activity in sheep brain. However, I think that Authors should consider the following points:

Major points:

1) According to the information provided by the manufacturer/distributor of the BDNF ELISA test the BDNF rapid ELISA kit used in this study has species reactivity for human, rat and mouse. As the samples used in this study originated from a different species (sheep) Authors should either validate the used test for sheep samples (and provide the validation results as supplementary material) or use a sheep BDNF ELISA test (which is available from multiple distributors).

2) Authors should correct the asterisks in the figures because the labeling of significance does not match with the results and the figure legends:

In Figure 1 a) the significance between the higher KYNA dose vs. control is P<0.001 (according to the results) which should have been labeled by 3 asterisks (according to figure legends) but in the figure there are 2 asterisks.  

In Figure 3 b) and d) the significance between the lower KYNA dose vs. control is labeled by 1 asterisk but the figure legend does not mention the meaning of 1 asterisk (which is P<0,05). Although, according to the results those significances are P<0,01 which should have been labeled by 2 asterisks.

3) Data are presented in mean±SEM but the Authors did not mention the exact numbers. The exact mean ±SEM values of the treatment groups should be added to the results.

Minor points:

1) In the conclusion the Authors mentioned that KYNA (or its derivates) may serve as a treatment strategy of certain neurological and psychiatric disorders. I think in the Discussion section the Authors should also mention the limitations of KYNA usage in a few sentences (e.g. 1.exogenous KYNA is not able to cross the blood-brain barrier; 2. elevated KYNA in the brain is also associated with mental disorders such as schizophrenia and depression: https://doi.org/10.3390/nu12051403).

Reviewer 3 Report

In this manuscript, authors injected kynurenic acid into the third ventricle of sheep brain, after 3 hours, hippocampus were harvested and detect the mRNA and protein level of BDNF, TrkB, OGG1..the repair activity of glycosylases such as 8-oxoG were also tested.

1. I noticed that author didn’t include blank control in experiment, control group in this paper is sheep received vehicle control. It is probably fine for mRNA and protein level of BDNF, there is low base expression. But in glycosylase activity test, the control activity is high, which might be caused by the brain trauma.

2. What’s the number of samples in each of your experiment? N =6 for all experiment?

Round 2

Reviewer 2 Report

I appreciate that the Authors considered my suggestions and made the required corrections and added the requested informations to improve the manuscript.